# Qualitative evaluation of a pilot mental health program for public safety personnel with post-traumatic stress disorder

Lucas M. Seuren[1], Arija Birze[1], Kathleen G. Dobson[2,3], Cameron Mustard[2,3], Walter P. Wodchis[1,4]*

1 Institute for Better Health, Trillium Health Partners, Ontario, Canada, 2 Institute for Work & Health, Ontario, Canada, 3 Dalla Lana School of Public Health, University of Toronto, Ontario, Canada, 4 Institute of Health Policy, Management and Evaluation, University of Toronto, Ontario, Canada

* walter.wodchis@utoronto.ca

## Abstract

Public Safety Personnel (PSP) such as police officers, firefighters, and paramedics are at an increased risk of post-traumatic stress disorder (PTSD) due to frequent exposure to potentially psychologically traumatic events. Return-to-work trajectories can be challenging, as treatment programs are often not tailored to PSP, leading to long-term disability. To better support PSP, the work disability insurance authority in Ontario, Canada commissioned a mental health treatment program for PSP who receive benefits for a diagnosis of PTSD. Staff in this program received specialized training by a credible PSP organization (Wounded Warriors Canada) for working with PSP. We conducted a formative evaluation of this pilot program. Qualitative, semi-structured interviews were conducted with service providers (n = 11) and PSP clients who had completed the program (n = 19). The aim was to understand whether and how the program met client needs, how it could be improved, and how this could inform other mental health treatment programs for PSP. Using descriptive qualitative methods, we found that participants reflected positively on the program's appropriateness, acceptability, and effectiveness. Credibility was a central concern across all three domains. Having a program that was exclusive to PSP allowed staff to tailor their treatment approach to unique PSP needs, which offered credibility as a PSP treatment program, and it made it possible for PSP to be in an environment where they felt safe.

## Introduction

Post-traumatic stress disorder (PTSD) is a major cause of disability among public safety personnel (PSP, typically including police officers, firefighters, paramedics, correctional workers, and emergency communications personnel) [1,2]. PSP experience a high burden of potentially psychologically traumatic events (PPTE) [3,4]

**Data availability statement:** Data contain potentially identifiable and sensitive patient information. Data can be made available by contacting the corresponding author or the Institute for Better Health Director of Research Operations, Delilah Ofosu-Barko (Delilah.Ofosu-Barko@thp.ca) upon reasonable request. The data from this project will be stored for the foreseeable future on secure institutional data repository on the Institute for Better Health data repository.

**Funding:** This work was supported by the Workplace Safety and Insurance Board (MUST2022 to CM and WPW). WPW is funded by a Research Chair in Implementation and Evaluation Science from the Trillium Health Partners Foundation. The funders had no role in the data collection and analysis, decision to publish, or preparation of the manuscript.

**Competing interests:** The authors have declared that no competing interests exist.

contributing to high rates of PTSD and other mental health challenges [4,5]. The symptomology of PTSD (e.g., hypervigilance, avoidance, insomnia, difficulty concentrating) impacts the ability of PSP to continue working [6], and PSP work involves continual re-exposure to PPTE, making return-to-work particularly challenging for those affected. Many face considerable barriers to treatment and a functional return-to-work process because of workplace stigma, poor understanding among mental health providers, and lack of support from employers and insurance boards [7,8]. Furthermore, there is a lack of tailored, evidence-based, specialized mental health support and treatment programs [2] and limited evidence of why interventions would be effective or not [9]. The perspectives of PSP on their treatment are rarely considered [3,10,11] making it difficult to develop support and treatment programs that are person-centered and tailored to PSP.

Within the current literature, there is a dearth of research that uses qualitative methods to understand the treatment needs of PSP, and even less is known about the experiences of PSP with existing and emerging treatment programs [3,8–10]. To our knowledge, no study has compared the experiences and perspectives of PSP and mental health providers on treatment. By providing this comparison, we aim to provide a comprehensive view of how PSP are best supported and to uncover discrepancies between the perspectives of staff and PSP. This is critical for developing programs that are tailored to the treatment needs of PSP and to increase acceptability by PSP.

### Mental health risks of PSP work and its impact on PTSD, disability and return-to-work

PSP are at a considerable risk of developing PTSD during their career [12]. Estimates of PTSD prevalence among PSP are considerably higher than the general public [4,13], ranging from 4% to 54%, varying primarily by profession, method of assessments, and geographical location [12,14–17], A 2018 exploratory study among Canadian PSP [5] found that 23.2% of respondents screened positive for PTSD based on the PCL-5 checklist (a validated self-report measure for assessing PTSD symptoms [18]). One potential cause of the high incidence of PTSD is frequent exposure to a range of PPTEs, what is called the Building Block effect [19]. A large survey study among Canadian PSP [4] found that participants reported having been exposed to on average 11 of 16 PPTEs of the DSM-5 Life Events Checklist (LEC-5). In comparison, a study found that the modal number of PPTEs was only three for members of the general public (which potentially includes PSP) [13]. PSP also reported frequent re-exposure (e.g., 90.6% of PSP were exposed to physical assault of whom 48.7% reported 11 or more events), which can cause an accumulation of occupational injury over time [4].

The frequency, intensity, and nature of PPTEs alone, however, do not fully account for the prevalence of PTSD among PSP. PSP typically work in an organizational environment where mental health challenges are stigmatized [3]. Moreover, many PSP feel unsupported by their organizations [10,20]. Early and ongoing intervention has been argued to be preventative of developing PTSD, but PSP are hesitant to reach

out to support services, and when they do – or when they are mandated to – services are generally found to be inadequate or untrustworthy [21]. Employee assistance programs and mental health education and training are often seen as checkbox exercises, and PSP feel they are not provided with applicable training to develop appropriate skills [22].

Moral injury, the psychological distress PSP may face when they are forced to act in a way that does not align with their values or where they are witness to such actions, has been proposed as a risk factor for PTSD [23–25]. Moral injury can also be caused by organizational approaches and treatment of employees – organizational or institutional betrayal – either where the PSP felt unsupported by their organization in their line of work or when experiencing difficulties with their mental health [21,26]. One study found that for PSP who had been medically discharged due to PTSD, the sense of organizational betrayal was more impactful than the PTSD itself [26].

### What does good PTSD treatment for PSP look like?

Systematic reviews on treatment for PTSD among PSP show that while there are studies on the effectiveness of various protocols, there is limited evidence on what makes these protocols work and for whom [9,14]. One reason PSP may be hesitant to seek out treatment is the perception that treatment modalities are not designed for them [8]. PSP have a unique cultural identity [10] due to the nature of their job, the organizational culture, their recurrent exposure to PPTEs, as well as the nature of these PPTEs. Therapists may be perceived as lacking the cultural competence to appreciate their trauma and support their recovery [11,12,21]. Where PSP seek out intervention, they may struggle with sharing their traumatic experiences, especially when it comes to varying identity characteristics such as gender [27].

Treatment for PSP should be culturally sensitive, meaning treatment providers need cultural competence and group treatment should be exclusive to PSP [7,8,10,11]. Providers require an understanding of the nature of PSP work as well as proficiency in the language of PSP to interact with them and to demonstrate cultural competence [21] including sensitivities around moral injury and institutional betrayals [25].

Building a therapeutic relationship requires not only the perceived ability to support PSP but also trust that the therapist is acting in the best interest of the PSP instead of an employer or insurance board [8,27]. This can be undermined in various ways. For example, PSP may feel that they are being pressured to return to work [7]. There may also be concerns with providers sharing information about PSP with disability benefit insurance providers or their employers: patient-provider confidentiality is core to cultural sensitivity [8,11].

### Study objectives

In December 2021, Insight Health Solutions launched a First Responder Mental Health (FRMH) pilot program for PSP with lost-time claims for PTSD attributed to occupational exposure from across the Canadian province of Ontario. Commissioned by the Ontario Workplace Safety and Insurance Board (WSIB), this was the first Ontario-based comprehensive mental health program for PSP. It is an evidence-based program, created in collaboration with partners (Wounded Warriors, EHN Guardians Gateway, Woodstock Hospital) who had extensive experience in trauma, anxiety and mood disorder assessment and treatment. All program staff received cultural competency training from Wounded Warriors Canada, a national mental health service provider delivering culturally informed services to trauma exposed professionals and their families.

In this study, we use an implementation science approach to understand the experiences of PSP and staff with this pilot mental health program. Implementation science is defined as "the scientific study of methods to promote the systematic uptake of research findings and other evidence-based practices into routine practice and, hence, to improve the quality and effectiveness of health services and care" [28]. Interventions are considered in a broader context to understand why they succeed or fail. Various theories and frameworks (e.g., Normalization Process Theory, Consolidated Framework for Implementation Research), have been developed to help researchers and implementers think through the process of implementation [29].

To understand the extent to which the FRMH program addresses the needs of PSP, we conducted a formative evaluation, guided by an implementation framework by Proctor et al. [30], which aims to conceptually synthesize other frameworks and theories. Contrary to summative evaluations, which evaluate the outcomes of an intervention, formative evaluations focus on the implementation of an intervention, with the aim of making recommendations for contextually sensitive adaptations [31]. As reported elsewhere, no association was found between program participation and return-to-work outcomes [32]. The objectives of this qualitative study were to understand the perspectives of participating PSP and providers on the program and to develop recommendations for improving the program. We addressed the following research questions:

1. To what extent does the FRMH program meet the treatment needs of PSP?

2. What can we learn from a formative evaluation of the FRMH program about how to better support PSP with PTSD?

## Methods

### Ethics statement

The study was considered quality improvement by the Trillium Health Partners Research Ethics Board and deemed exempt from formal review. Participants were recruited between August 2023 and March 2024. All participants provided written informed consent to participating in the study and to their interview data being recorded and used for research and in publications. The research team had no affiliation with the program development or implementation.

### Study design

We conducted a qualitative descriptive [33,34] study using semi-structured interviews to investigate the implementation of the FRMH program using the conceptual framework for implementation outcomes developed by Proctor et al [30]. Qualitative description allowed us to stay close to the data while using the Proctor et al.'s conceptual framework to guide our interpretations of the data [35]. By combining perspectives of PSP and treatment staff, we could identify complementary as well as discordant views.

We used the Proctor et al. framework [30], as it allowed us to differentiate between potential implementation failures and intervention failures. As the FRMH program was the first dedicated program for PSP in Ontario, we decided to focus on the acceptability and appropriateness of the program for PSP. Furthermore, to assess whether and how the program adequately supports PSP with PTSD, we also focused on effectiveness. For this study, we defined these concepts as follows:

• *Acceptability*: The extent to which the FRMH program was viewed as favorable for and by PSP.

• *Appropriateness*: The perceived fit of the program to the treatment needs of PSP.

• *Effectiveness*: The extent to which the FRMH program was perceived to support the treatment goals of PSP.

The COREQ checklist for qualitative research [36] was used to help provide rigor to the study and the transparency to the final report.

### Setting and context

The FRMH Program is a treatment program for PSP diagnosed with PTSD living in Ontario, Canada. The program consists of two main modalities: a high-intensity outpatient (HIT) program and a residential program. Before admission to either modality, PSP undergo a comprehensive assessment. Potential outcomes of this assessment can be a recommendation to continue with treatment in the community or to be admitted to the residential or HIT program.

Our focus in this study is on the HIT program, as it had been newly developed and was still in its pilot stage at the time of the formative evaluation. The program lasts 14 weeks, with an optional 10-week period of aftercare. Treatment modalities include psychotherapy (primarily cognitive processing therapy (CPT), sometimes complemented with dialectical behavior therapy (DBT), both individually and in groups), psychoeducation and skills groups, medication management and reconciliation, a physical reactivation program that supports patients with incorporating movement and exercise into their recovery, and return-to-work services. PSP fill in DSM-V symptom questionnaires every two weeks. PSP are members of a small cohort of 3–7 PSP who participate in group therapy, psychoeducation and physical reactivation sessions together. All sessions are conducted remotely using secure videoconferencing software. PSP can be referred to the program by their WSIB case manager or they can self-refer. Treatment is confidential and no information is shared with employers. Treatment staff do provide written progress reports for WSIB case managers.

## Participants and recruitment

Participants were recruited using self-selection sampling. All treatment staff involved in the program and all PSP who were discharged from the program over the 9 months of July 2023 to March 2024 (formal end date of the evaluation project) were invited to participate. An administrative member of the program staff sent recruitment emails to all PSP upon their discharge from the program. Invitations included a brief introduction to the project and direction to contact A.B. should they be interested in sharing their experiences with the goal of informing changes in future cohorts. After addressing any questions and concerns, all PSP who expressed an interest in participating then completed an interview.

A member of the research team (L.S.) presented the study to therapeutic staff delivering the program at a team meeting, where staff had an opportunity to ask questions about the study. An invitation letter was subsequently distributed via email by the program manager. Two months later, personalized emails were sent to all staff who had not responded to the initial invitation. After completing interviews with therapeutic staff, invitation letters were sent to staff conducting comprehensive assessments of potential patients prior to the program.

## Data collection

Semi-structured interviews were conducted and audio-recorded over Zoom, with researchers taking notes throughout to help guide the interviews and analysis. Two standardized interview guides were developed, one tailored to PSP and one to mental health staff. Interview questions were designed by drawing on Proctor et al.'s framework [30] to probe participants' perspectives on the appropriateness, acceptability, and effectiveness of the FRMH program (see S1 File). L.S. developed initial drafts of the guides which were then revised based on iterative feedback from the research team. We did not validate the guides with PSP or mental health staff. Treatment staff were interviewed by a researcher with expertise in qualitative and mixed-methods research on implementation science and healthcare service delivery (L.S.). PSP were interviewed by a researcher with expertise in qualitative and mixed-methods research on occupational and posttraumatic stress in PSP as well as lived experience of PSP culture as a 30-year partner of a long-serving active PSP (A.B.).

All audio-recordings were transcribed by an external transcription agency. Transcripts were reviewed for accuracy and anonymized by L.S. and A.B. and then imported and analyzed in NVivo 12.6 Plus.

## Analysis

Data were analyzed through an inductive approach, using effectiveness, appropriateness, and acceptability as sensitizing concepts. After reviewing transcripts, two team members separately coded the interview transcripts of the therapeutic staff (L.S.) and PSP (A.B.) using an open coding approach. Coded data were then analyzed through the lens of PSP and staff perspectives on what shaped acceptability, appropriateness, and effectiveness of the FRMH program. Open codes were reviewed and grouped according to sensitizing concepts. Preliminary themes were then developed for each of the

three main categories, and these themes were subsequently discussed and refined through several small team meetings (L.S. and A.B.). Agreement on all themes was sought and achieved through discussion. Care was taken to identify complementary as well as discordant views of PSP and staff on key concepts (negative case analysis). When themes were only derived from one group of participants, these were included if they provided vital perspectives to understanding and improving the program. Analytic saturation was assessed when agreement was reached that all codes were adequately captured within preliminary themes. Final themes were developed by L.S. and A.B and reviewed, discussed and approved by the larger team.

## Results

Interviews were conducted with 11 service providers (seven in HIT treatment and four in comprehensive assessment, response rate 65%), and 19 PSP (response rate 19%, see Table 1). Interviews with staff lasted on average 53.3 minutes (range 40–90 minutes, median 48.9 minutes). Interviews with PSP lasted on average 79.5 minutes (range 52–150 minutes, median 77). In the rest of this section, we discuss the perspectives of staff and PSP around the appropriateness, acceptability and effectiveness of the FRMH program.

## Appropriateness

One core finding was that *credibility* of both the program and its staff was a core concern for PSP. Credibility here can be understood as the evidence that the program was designed for PSP: it provides trust, legitimacy and cultural safety. Both PSP and mental health staff reflected that the design of the HIT program was a good fit with the unique treatment needs of PSP, both in terms of its programming (e.g., CPT, psychoeducation) and its overall structure (e.g., virtual, high intensity, cohort). However, the HIT program was not adequately tailored for PSP with comorbidities (e.g., suicidality, alcohol use disorder), making it less appropriate for those struggling with mental health concerns additional to PTSD..

**Exclusive to PSP.** Both staff and PSP participants felt that what set the program apart from other mental health programs or community therapy was that it was exclusively for PSP. Staff explained that by restricting participants to PSP, the program could also be tailored to their unique treatment needs. The value of this approach was validated by PSP. Many participants felt their unique professional circumstances around their PTSD were being attended to.

*I think that in itself [that it is a first responder-specific program] is very unique to the program and makes it more desirable. So, knowing that that's the only clientele they're dealing with, gives a little bit of comfort knowing, OK, they might not be able to relate to what I've gone through. But they're hearing it from so many different people, that they might have a better understanding or empathy for it. (PSP011, police officer)*

**Table 1. Demographics of PSP participants.**

| Profession | | |
|---|---|---|
| | Paramedic | 8 |
| | Police officer | 6 |
| | Mixed (fire, communications, paramedic) | 2 |
| | Corrections | 1 |
| | Police communications | 1 |
| | Nurse | 1 |
| Gender | | |
| | Male | 11 |
| | Female | 8 |

Similarly, a paramedic explained how the unique PSP-specific knowledge of the staff helped to make them feel understood:

> I think it's so suitable for us, because they dive into stuck points that only first responders would have. It's so specific to us in the way we think, in the way we act at work, that it wouldn't make sense for the general public. (PSP017, paramedic)

**Groups provide validation.** As the program was exclusive to PSP, both PSP and staff found group sessions helped *provide validation* and *overcome isolation*. PSP consistently discussed the profound impacts of the presence of other PSP and the opportunity to share their experiences and offer help among the safety of trusted others. They expressed relief that they could finally identify with similarly affected peers, sometimes for the first time.

> Having our feelings and experiences validated by hearing someone else describe events, or feelings, or complete inabilities that we also experienced made us feel we weren't alone, and made us feel, OK…other people are out there going through this…I'm not fucked up…This is something that happened to me, this isn't who I am. (PSP027, corrections worker)

Staff endorsed these PSP perspectives, explaining that group sessions provided PSP with safety and validation, and they could build relationships and trust with PSP in their cohort during the program. Staff felt that in the PSP work environment, mental health problems can be isolating as there is a need to be, and be seen as, mentally impervious. Going through a program with a cohort of PSP normalized the impact of their injuries and their coping strategies, and it made them feel more comfortable sharing their experiences.

> They're able to just see that they're not alone in their experience...that experience of having PTSD as they report is just so isolating, and they're confused by what's going on. And they think that their mind is broken forever. Seeing other people that have been in similar situations can help to challenge those beliefs in a much more powerful way than a clinician supporting them and changing the beliefs. (Clin010)

**High intensity therapy provides needed structure.** Having a high-intensity, structured program was seen as supportive of the unique needs and organizational culture of PSP: it provided direction and purpose. Although the renewed routine and several weekly appointments were an adjustment, the majority of PSP felt they were ready for the program and appreciated the structure, routine, and accountability it was bringing back to their daily lives. Moreover, the focus and attention on the intervention and program homework filled their time in new and productive ways.

> [A]t first, it was a lot, and I was overwhelmed, but then it became my routine and I loved it, and I kind of needed it, I needed that routine, that structure. Even the socialization, I needed that, it was good. (PSP017, paramedic)

Staff also recognized that the intensity could be overwhelming at the start, but they felt it was needed to make progress with trauma recovery. The lack of intensity in community therapy was seen as precisely why PSP often did not make much progress. Furthermore, because PSP experience a high-pressure work environment, staff felt the program functioned as something like a training camp: if PSP could cope with the intensity of the program, they were more likely able to cope with the daily structure of work.

> I think having a structured program like that can be very helpful, because a lot of the clients, they have a struggle with structure and daily routine. And this program can help them to kind of start building that daily routine. (Clin015)

**Virtual delivery provides safety.**  PSP and treatment staff felt a virtual delivery made the program accessible to PSP across Ontario, and that it was a good fit with the trauma experiences of PSP. Many patients were self-isolating at home to some degree and regularly experienced anxiety related to leaving or travelling from their homes. Online therapy made attending the program possible and gave patients the opportunity to engage in ways different from in-person appointments. The virtual meetings allowed PSP to be vulnerable from the comfort and safety of their own homes because of the felt distance inherent in video clinics.

*So for me, the virtual piece actually made it easier to be more vulnerable because … I feel like I have this safety net because they're not actually physically there watching me cry and break down. (PSP011, police officer)*

While many PSP found virtual necessary to access therapy, others described a lack of connection to their cohorts due to the virtual delivery. Staff also saw it as a trade-off between quality and accessibility. They had not received training for virtual therapy, and most had learned through experience during the COVID-19 pandemic. With virtual therapy, it was not possible for PSP to make the kinds of lasting connections that they would with in-person groups.

*In group therapy, when it happens in person, we don't have control over…when they're exchanging numbers, they're meeting outside for coffee, maybe they build friendships. So that is a big piece, because a lot of these folks are so, they're so isolated …they cannot talk to anyone who can resonate with them. (Clin014)*

**Tailoring to specific needs and co-morbidities.**  Staff reported that one of the goals of the comprehensive assessment was to provide appropriate treatment recommendations for potential patients. PSP with more severe symptoms, maladaptive coping strategies (e.g., excessive alcohol use), and those at risk of harming themselves or others were recommended to the residential treatment program. However, there were various reasons why PSP would or could not take up this recommendation (e.g., family responsibilities). This created a problem for the treatment staff who felt they were unable to provide the right support for PSP. Even where staff felt they had the skill to address coping strategies, treatment sessions were focused on trauma processing and so addressing comorbidities would take time away from that.

*It's hard, especially for me, to do so much work within five sessions. Especially when we're talking about alcoholism, substance use, harm, which are usually very ingrained traits after years of those behavioral patterns. (Clin016)*

Similarly, some PSP respondents perceived the program as underprepared for treating PSP with complex needs such as suicidal ideation. As the high-intensity program was not designed to support PSP with alcohol use, suicidal ideation or other conditions, PSP were careful with whether they would recommend the program to their colleagues.

*I'd say, "You may want to try this program, you work out with a kinesiologist and individual therapist and assessment, medication." But if I know someone's struggling with suicide and self-harm and drinking, then I'd be no…they're not equipped to deal with higher priority patients with, they call it challenging behaviors, which is code for self-harm and drinking. (PSP015, paramedic)*

Finally, while the program was designed to meet the unique needs of PSP, and there was a recognition among staff that PSP have "cumulative trauma that they go through in their occupations" (Clin014), some PSP found that the program was overly focused on singular incident exposure, making them feel the program was not for them.

*Very early on I realized that there was going to be a problem…because it wasn't a singular incident that brought this on, it was accumulation. All the work, like the worksheets, and all the things, and even the PTSD questionnaires, didn't*

*reflect that, it was like, all the questions were geared to singular events….I was like, "OK, well this isn't geared to me."* (PSP026, paramedic)

## Acceptability

Securing buy-in of PSP for the program depended on the fit with their treatment needs and whether the design and implementation of the program were done in a way that they felt worked for them. Staff reflected that the program and staff needed to build credibility as a PSP-specific program. PSP had more extensive concerns, particularly around support in three areas: communication of program details and expectations; group cohesion and composition; and the discharge process.

**Building credibility.**  One core concern in engaging with the FRMH program was the extent to which PSP felt the program had been designed for them, thereby signaling credibility. Treatment staff needed to have an appreciation for the unique work experiences and organizational culture of PSP. Staff felt that the Wounded Warriors program helped them understand the work experiences of PSP, what kinds of situations they were repeatedly exposed to, and what their operational and organizational stressors were. However, some PSP felt that staff were still not fully prepared to address their needs. While staff were always perceived as well-intentioned, clients felt their lack of understanding required extensive explanation and prevented deeper engagement.

*Out of all the contact I had, I would say maybe one, I felt like had experience and knowledge and knew… I felt like a lot of them had just finished school and had just graduated, and didn't necessarily have experience with first responders or with…our kind of world. (PSP011, police officer)*

At times, differences in identity between PSP and staff (e.g., age, gender) also made it hard for PSP to share the most intimate details of their traumas. A police officer stated the clinicians he worked with reminded him too much of his own daughter, which made him hesitant to share what he was really struggling with. He suggested that if there was a way to train and implement the program with police officers then he would be able to "make that physical connection with someone who's got street credibility" and be "engaged in the program a lot better" (PSP019).

Staff also raised the tension with a return-to-work program funded by an insurance board. It can signal to PSP that the program is mandated or that the program is more aligned with the interests of the insurance board and employers. Staff felt that this could undermine trust. Some PSP echoed this: they were unsure of who had access to their symptom profiles, making them less forthcoming.

*I honestly thought that those were going to my employer too. So, I didn't want to talk about suicide ideation, I didn't want to talk about, that I was feeling negative and pissed off and angry. Because I didn't know where those questionnaires were going. (PSP019, police officer)*

**Need for improved program communications.**  PSP shared frustrations over the lack of clear and consistent communication from the program. This impacted their satisfaction and confidence through reduced opportunity to prepare and feel engaged ahead of time. One police officer, after completing an inpatient program prior to beginning the HIT program, described an isolating waiting period that eroded progress and confidence in the program:

*So, when I left [my inpatient program], it was six weeks before anybody contacted me, and I'll be honest with you, I lost a lot of momentum. I felt alone, I felt isolated, I felt like, "OK, what's happening?" I called [the program] multiple times to say, "What is happening?" they didn't even know who I was. (PSP019, police officer)*

Other PSP also struggled with organizing and making their way through intake materials. They felt clear communication and explanation over the process and contents were necessary. One police officer recalled filling out required surveys and suggested that, at least initially, someone could explain the questions so participants fill them in correctly.

PSP also wished for more consistent communications when there were changes to programming scheduling and staffing. The lack of communication in such circumstances created feelings of uncertainty and anxiety over an inability to prepare for inevitable disruptions or changes.

*They brought in two brand new clinicians for one of the groups that nobody had ever met…It is that important to some-one like me, because my whole world starts falling apart over the simple fact that I didn't have the same clinician, right. And, but providing me with enough information…talking about what it will look like to have this new psychiatrist speak to you, and how do you feel about that? And how are you going to prepare for it? Any of those things would be helpful. (PSP011, police officer)*

**Challenges with group cohesion and composition.** While group sessions provided a safe space for listening and sharing as well as validation of their experiences with PTSD, some PSP felt that they would be improved through additional group facilitation and engagement strategies. They described an inherent cohesion in the PSP community and felt that those bonds needed further facilitation alongside explanations of how meaningful engagement in groups may enhance effectiveness.

*You need to create a team environment and create that bond …we all do have that sense of family and cohesiveness, just because we are first responders…But it's hard to facilitate that in an online group…of 12 individual strangers that you know nothing about. (PSP025, paramedic)*

Some PSP felt the contribution of peers was invaluable for understanding and processing their traumas. When interactions were limited, PSP felt the need for more time and opportunity to work as a team. One fire/paramedic PSP exclaimed:

*"I think there absolutely 100% needs to be more involvement with the stuck points and how to deal with the stuck points, and doing it as a team." (PSP018).*

Others suggested that the cohesiveness and the efficacy of their group sessions could be improved with more consideration for group composition. Their suggestions included gender, length of time off work with a mental health injury, seniority or rank, and the nature of their traumatic exposures. One PSP leader described how their seniority prevented them from fully engaging and led them to feelings of isolation:

*"[O]ne of the biggest stumbling points was being identified as being in management, a lot of the triggers for the other participants were specifically related to management, so I was kind of isolated…from the group." (PSP023, paramedic)*

In another instance of reflection on group composition, one PSP described her experience as one woman in a group of men as preventing her ability to fully relate to the experiences of others:

*The only critique is that I couldn't relate to these other men, one because they're a lot older, they're all men, and they all have wives and children. And that was something I found difficult (PSP017, paramedic)*

**Lack of a patient-centered discharge process.** PSP often felt as though they were discharged abruptly. Many suggested a more comprehensive transition was needed to prevent erosion of progress and support continued healing.

The lack of warning and abrupt end compromised the sense of purpose and direction they had established, and it was counterintuitive to the high-intensity structure of the program.

*I would have just liked kind of a warning, "OK, we're coming to the end now, so this next week will be our final session, and this is what's next." Whereas it kind of just came to a very blunt end…I just really need some sort of direction… someone to help me figure out how, what my next steps are, what I need to do now. (PSP017, paramedic)*

One of the biggest difficulties for some PSP was the realization that they had completed the program but were "not well still" and did not have a clear plan forward. One police officer found the discharge process both abrupt and dehumanizing:

*After I finished the aftercare, it was like "OK, bye, we're done." Done…I didn't hear from anybody after that. And I was like, "OK, so I'm just kind of what, like left on my own now? I'm not OK." So what do you do with people who are finished this timeline, but actually are not well still, you just leave them?...So that I felt really like…"OK, well I don't matter. I'm just like a number." (PSP011)*

**Effectiveness**

The interviews with staff and PSP indicated that effectiveness could be understood in different ways. Respondents felt the metrics that the program uses to track recovery, such as patient progress on clinical questionnaires, did not provide a comprehensive view of its benefits. Moreover, we found that the goals of the program as defined by WSIB and program leadership (e.g., ready to start return-to-work trajectory) could be at odds with the goals of the treatment staff and patients (e.g., improvement in daily functioning). These different conceptions of effectiveness meant that how the program was seen to support PSP varied considerably across interest groups.

**Defining treatment goals.** As the program was commissioned by WSIB, its main aim was to support PSP with their return-to-work trajectory. However, staff did not expect that upon completion of the program PSP would be ready to return to work in all cases. In fact, focusing on return-to-work as *the* goal could be detrimental to recovery. PSP expressed feelings of anxiety over returning to work as they approached the end date of the treatment program, shaping their responses on the questionnaires.

*So, once I kind of got halfway through the program…my overall general sense of anxiety went down significantly once I started the program…but because I felt like there was a certain level of expectation that once I was done the program…I would be right back into work…So, for me, there was this uncertainty, grey area that was just kind of looming at the end of the program. (PSP020, paramedic)*

PSP suggested there were additional reasons why the questionnaires were not necessarily reliable tools for assessing progress. One police officer explained that as they moved through the program, their understanding of trauma and its symptoms grew. This improved their ability to distinguish and recognize symptom patterns and more accurately fill out the questionnaires, which could give the appearance of an increase in symptoms or severity.

*I've been able to better identify through doing the surveys, based on what I've learned and what I know now … originally, I was kind of hitting lower down on the PTSD scale. But that's because I was just kind of lumping everything together saying, "Oh…it's just stress or it's just anxiety." (PSP022)*

Other PSP were aware that they would be returning to the very environment where they incurred their injuries. When they felt that the organizational conditions may not have changed, this left them feeling unsupported and unsafe. Some described this as a moral injury with lasting impacts, especially with respect to their readiness for return to work.

*We want to hear from our bosses…you want to know that you're valued and you're appreciated. And there is none of that…it's horrible…I've been involved in…[several] major events…and never got a call, never got…a "How you doing?"…my workplace failed to support me…and they're doing it to this day. (PSP019, police officer)*

For treatment staff, effectiveness was therefore a highly individualized concept. Their treatment focus was on daily functioning and quality of life. While staff recognized that improvements in these areas could help patients with a return-to-work, they felt that if they put the focus on return-to-work itself, this would be counterproductive. Staff found that they could serve patients best by focusing on their individual needs and goals around symptom improvement.

*I don't care if you're back to work or not. For me, the purpose of the program is to get you better, functionally better, right, or to improve your symptoms, if that leads to work, then great. (Clin016)*

Staff considered symptom improvement to be in service of returning-to-work: a vital part of a meaningful life. Supporting patients with an improved quality of life and daily functioning therefore meant talking about return-to-work and potentially helping patients get ready to at some point enter a return-to-work process, but managing these conflicting goals was difficult.

**Understanding PTSD and learning coping skills.** Many PSP shared that when they first entered the program, they didn't really know what trauma was or why and how they might have developed a posttraumatic injury or understand its lasting impacts. Notwithstanding this general lack of awareness and knowledge around the potentially traumatic impacts of their work, PSP suggested the program helped them better understand trauma and its impact and it taught them new skills for coping with their various symptoms.

*I have felt that I was missing something, that I didn't understand myself…I just felt this floundering, I kept trying and doing everything, but nothing seemed to kind of connect. And I felt like I needed a program to…tie everything all together, and that's exactly what it has done for me. (PSP020, paramedic)*

PSP also consistently reported how much they valued the various coping skills they learned within the context of their traumatic experiences, throughout the intervention. As one police officer described their new awareness around strategies for coping, we can also see their increased understanding of the nature and complexity of living with trauma:

*I think a lot of the techniques I learned were very, are very useful…I just wanted to get better, but I never knew that there were coping techniques. I never thought – in my head I'm like you just get better, there's no in-between….like breathing techniques I never even heard of that before." (PSP011)*

## Discussion

This formative evaluation of a PSP mental health program in Ontario provides valuable insights into how mental health treatment programs could be designed to better support PSP with PTSD. Using the concepts of appropriateness, acceptability, and effectiveness from Proctor et al. [30], we found that credibility, both of the program and the treatment staff, was a core driver among PSP acceptance of and engagement with the program. PSP have unique workplace and trauma experiences. Tailoring treatment programs to the specific needs and experiences of PSP by focusing on practical skills that help them cope and build resilience is critical for credibility and to provide a safe space [3,8,12,27]. With careful consideration of group composition (e.g., gender, rank, profession), the cohort model and psychoeducation helped normalize and destigmatize PTSD and the other mental health challenges, that keep PSP from returning to work [3,7,37,38]. Group therapy can help profoundly validate PSP

challenges and secure buy-in by offering a safe space where they recognize that others have similar challenges with work-related trauma.

Our study also indicates that effectiveness of treatment is complex. While the focus on recovery and return-to-work is understandable from the perspective of an insurance board, many PSP are more practically oriented, wanting to better understand PTSD and the other learning skills that can improve their self-efficacy, even if that does not immediately lead to better scores on symptom questionnaires. A better understanding of trauma recovery and return-to-work is nonetheless necessary. Edgelow et al [39] found that of the PSP claims filed in Ontario from 2017 to 2021, only 52.7% led to a successful return-to-work [32]. A quantitative evaluation of the FRMH program found that among PSP who were enrolled in the program between November 2021 and June 2023, 29.9% had ceased receiving loss-of-earnings benefit by November 2024, a proxy for return-to-work. This compares to 32.5% of those who were referred to the program but did not enroll in the treatment program (although these were not matched cohorts). Both are considerably lower than the number of PSP who were not referred before returning to work (41.9%) [32]. While referred PSP differed substantially from the study population of PSP with PTSD (e.g., longer claim durations, different occupational distribution), these findings do indicate the difficulty in providing effective treatment and support in the PSP population.

**Integration with existing literature**

Our findings build on existing research on mental health treatment for PSP in three ways. First, PSP participants complained of a lack of continuity of care (CoC), noting the "abrupt discharge" from the program and the feeling of being left on their own. There is some evidence that a lack of CoC can be detrimental for treatment outcomes [40]. For patients with severe mental illness, CoC has been associated with lower suicidality, lower healthcare usage, and higher quality of life, albeit not with improved symptoms and functioning [41]. Our study similarly suggests that at least some participants noted a regression, due to what they feel is poor support when transitioning out of the FRMH program, which may have contributed to the lack of positive effect of the program [32].

While virtual therapy has grown since COVID-19, few studies have focused on group virtual therapy. The lack of co-presence raises questions about the type and quality of connections that PSP make, both with the therapist and each other, and how this shapes their ability and willingness to share and participate. While virtual therapy may not affect the therapeutic alliance compared to in person therapy [42], research does not address dynamics and participation in group therapy. PSP with PTSD are seen to be isolated, but as our study shows, video may not entirely address this deficit as there are no opportunities for informal connection. Video technology is designed for one-speaker-at-a-time, and so before or after therapy there are no opportunities for participants to chat in spontaneous small groups or one-to-one.

Finally, maladaptive coping strategies among PSP with PTSD such as alcohol use or self-harm and suicidality are well identified [12,43], yet our study shows these were not necessarily adequately addressed in the outpatient program, undermining program effectiveness. Inability of PSP to attend recommended inpatient programming who then opt for outpatient programming presents considerable challenges for the delivery of adequate outpatient treatment. One recent systematic review indicated that one in four PSP engage in hazardous drinking and one in ten engage in harmful drinking [44]. And while the association between PTSD and suicidality in PSP is poorly understood [45], PSP are known to be at an increased risk of suicidality [45,46], particularly if they are already experiencing poor mental health (e.g., anxiety, depression) [47,48]. Furthermore, moral injury among PSP may play a considerable role in both the experience of PTSD and suicidality, indicating the need to address moral injury alongside PTSD [49]. Finally, research among PSP has focused particularly on alcohol use as a comorbidity of PTSD [50,51], whereas our study shows that PSP may experience a range of comorbidities that could affect treatment outcomes, including eating disorders and compulsive exercise. Recent studies with military veterans and firefighters suggest that transdiagnostic treatment protocols may be required [52,53].

## Recommendations for practice

Treatment programs should provide flexibility to support PSP by presenting PTSD as the outcome of both singular and/or cumulative exposure to PPTEs. Some PSP found that treatment was designed around the traumatic impacts of singular events, but felt that it was the continuous exposure to a variety of PPTEs, making it difficult for them to identify particularly salient events, known as the Building Block effect [19], that led them to develop PTSD. Tailoring programs to accommodate this cumulative nature might improve program acceptability, appropriateness, as well as treatment effectiveness.

Second, cultural competence of treatment staff could be improved by either recruiting (and training) former PSP to or by having treatment staff going on ride-alongs or other means of familiarizing them with PSP work. Cultural competence of therapists has consistently been recognized as a missing link in many treatment pathways for PSP [7,8,11]. Having a therapist who does not know what their job is like can make PSP skeptical that the therapist can support them [54]. While the Wounded Warriors program was seen to be beneficial, PSP still found it difficult to connect with treatment staff in some cases. Demonstrating experience could improve credibility and thereby engagement.

Third, improved continuity of care and transitions from one point of care to another may protect against erosion of progress and support confidence as PSP enter and move through treatment and return-to-work processes.

Fourth, careful consideration of group composition that takes into account the social dynamics of gender, profession, rank, and nature of workplace exposures and PTSD can help to support better engagement and facilitate meaningful group cohesion, potentially boosting effectiveness.

Finally, although the program had an optional period of aftercare, many PSP felt that discharge was abrupt, and others reflected that their symptoms worsened due to concerns about discharge and return-to-work. Follow-up with PSP after discharge (e.g., 6–12 months) could help assess how specialized programs can better support the transition to either return-to-work, community therapy, or other support mechanisms.

Formative evaluations provide a valuable tool for continuous improvements of treatment programs. Findings from this study were used by FRMH program leadership to make adjustments to make the program more person- and PSP-centered. Content in the psychoeducation was revised and the size of groups was revised based on clinician feedback, the evaluation findings, and anecdotal feedback from clients. A psychotherapist with public safety employment experience (retired from policing) was also onboarded to support the lived-experience piece and credibility.

## Strengths and limitations

To our knowledge, this is the first study to combine interviews with PSP and mental health staff to investigate the treatment needs and experiences of PSP with PTSD. This allowed us to better understand how programs can be tailored to PSP, and why there may be potential mismatches between the treatment needs of PSP as reported by PSP compared to how they are perceived by mental health staff.

The findings of this study also have some limitations. We did not interview PSP who had chosen not to proceed with the FRMH program. It is to be expected that PSP who do not participate in the program have specific concerns that we missed and that could be critical for developing and implementing effective mental health treatment programs. Further bias may result from self-selection sampling. We also only collected limited demographic information (gender and professional role), meaning we cannot assess how representative participants were of the population [32]. While analytic saturation was achieved within the collected data, some findings were based on single interviews. Variability in potentially relevant participant categories (e.g., demographics, experience, profession) means that wider recruitment might have produced additional insights.

We did not distinguish in our analysis between the different PSP professions. Our analysis indicated that different PSP have different experiences (e.g., differences in the nature of the work meant that PSP would have different types of PPTEs), which can mean that a more granular analysis is required. This fits with previous studies that have argued that PSP as a category is too heterogeneous due to differences in the nature of the work [3,12].

Finally, it would have been valuable to have PSP or a mental health practitioner as part of the research team to design the study, interpret the results, and make appropriate recommendations. Lived experience of PSP is a central concern for credibility of treatment and potentially had similar impact on credibility of the research team. While the researcher who conducted interviews with PSP had experience with trauma-based research with PSP and lived experience of PSP culture as a 30-year partner of an active PSP, PSP may have been hesitant to participate in the evaluation without a PSP team member. Similarly, demographics like gender and age were found to be important for PSP to feel comfortable sharing their experiences in group therapy, which likely means some might have been less forthcoming if they perceived discrepancies with the interviewer.

### Future research

Based on our findings, we uncovered two areas where further research would be valuable. Our findings suggest that some PSP experienced improvements, but lacking quantitative data, we cannot draw definitive conclusions. Responses to symptom questionnaires (e.g., PCL-5) provide only a limited perspective: as they learn, PSP change their understanding and thus how they fill in these questionnaires. Even where they don't improve, PSP may still learn new skills and information about trauma-related injuries, thereby increasing their self-efficacy or confidence in managing their mental illness.

Second, the present program focused on *treatment* whereas additional focus on *prevention* is warranted. There may be opportunities for improved resilience and coping programs including education and workplace training to improve acceptance of traumatic events, an understanding of trauma-related injuries, and the development of appropriate coping and support, not to mention broader structural and organizational improvements that can be made to better support members. Testing and evaluation of such programs for PSP, which may have benefits in prevention, recovery and return-to-work, is recommended.

Third, many PSP in our study experienced maladaptive coping strategies such as alcohol use disorder, eating disorders or over-exercising. While there is research on the prevalence of AUD among PSP [12,43], nothing is known about the prevalence and severity of other coping strategies. Given the impact these strategies have on quality of life and what would be appropriate and effective treatment, studies are urgently needed that investigate the prevalence of additional maladaptive coping strategies and appropriate treatments.

### Conclusion

Specialized mental health treatment programs are potentially appropriate, acceptable, and effective for public safety personnel living with PTSD. By creating a therapeutic environment that is exclusive to PSP and where PSP can be in groups with those with similar backgrounds (e.g., rank, gender), programs can be tailored to their experiences and treatment needs, while also building credibility among PSP that the therapists and the program are acting in their best interests. Although there are ongoing efforts to destigmatize mental health, many PSP continue to be hesitant to seek treatment as doing so is seen as a weakness. While there is growing attention to prevention of PTSD by providing more supportive work environments (e.g., trauma training and education, destigmatizing mental health, psychological support from employers, timely access to mental health services) [21–23], specialized treatment programs, tailored for cultural safety, are urgently needed to support PSP with trauma recovery and return-to-work. High-quality treatment programs are crucial not only for PSP themselves, but for the wider public, which has a strong interest in a mentally healthy and resilient PSP workforce.

### Supporting information

**S1 File. Interview guides.**
(DOCX)

## Acknowledgments

We would like to thank all participants for their time and contributions and Leslie Vesely and Basak Yanar for insightful comments at various stages of the study and on earlier drafts of this manuscript.

## Author contributions

**Conceptualization:** Cameron Mustard, Walter P. Wodchis.

**Data curation:** Lucas M. Seuren, Arija Birze.

**Formal analysis:** Lucas M. Seuren, Arija Birze.

**Funding acquisition:** Cameron Mustard, Walter P. Wodchis.

**Investigation:** Lucas M. Seuren, Arija Birze.

**Methodology:** Lucas M. Seuren, Cameron Mustard.

**Project administration:** Lucas M. Seuren.

**Supervision:** Walter P. Wodchis.

**Validation:** Lucas M. Seuren, Kathleen G. Dobson, Walter P. Wodchis.

**Writing – original draft:** Lucas M. Seuren, Arija Birze.

**Writing – review & editing:** Lucas M. Seuren, Arija Birze, Kathleen G. Dobson, Cameron Mustard, Walter P. Wodchis.

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
