## [Decision Letter · Decision Letter 0]

22 Oct 2025

PMEN-D-25-00363

Qualitative evaluation of a pilot mental health program for public safety personnel with post-traumatic stress injury

PLOS Mental Health

Dear Dr. Wodchis,

Thank you for submitting your manuscript to PLOS Mental Health. After careful consideration, we feel that it has merit but does not fully meet PLOS Mental Health’s publication criteria as it currently stands. Therefore, we invite you to submit a revised version of the manuscript that addresses the points raised during the review process.

We look forward to receiving your revised manuscript.

Kind regards,

Lambert Zixin Li, Ph.D.

Academic Editor

PLOS Mental Health

Journal Requirements:

Additional Editor Comments (if provided):

Dear authors,

Thank you for submitting your manuscript to PLOS Mental Health. Your research addresses an important and timely topic.

We have invited several reviewers with relevant expertise to evaluate your submission. Many of them have provided detailed and constructive feedback aimed at strengthening the scientific validity and clarity of your manuscript. We kindly ask that you address all reviewer comments in a revised version of your manuscript and provide a point-by-point response, with particular attention to the feedback from Reviewers 1, 2, and 5.

We look forward to receiving your revised submission.

Sincerely,

Lambert Zixin Li, PhD

Reviewers' comments:

Reviewer's Responses to Questions

**Comments to the Author**

1. Does this manuscript meet PLOS Mental Health’s publication criteria?

Reviewer #1: Yes

Reviewer #2: Yes

Reviewer #3: Yes

Reviewer #4: Yes

Reviewer #5: Yes

Reviewer #6: Yes

2. Has the statistical analysis been performed appropriately and rigorously?

Reviewer #1: N/A

Reviewer #2: Yes

Reviewer #3: Yes

Reviewer #4: Yes

Reviewer #5: N/A

Reviewer #6: Yes

3. Have the authors made all data underlying the findings in their manuscript fully available (please refer to the Data Availability Statement at the start of the manuscript PDF file)?

Reviewer #1: No

Reviewer #2: Yes

Reviewer #3: Yes

Reviewer #4: Yes

Reviewer #5: Yes

Reviewer #6: Yes

4. Is the manuscript presented in an intelligible fashion and written in standard English?

Reviewer #1: Yes

Reviewer #2: Yes

Reviewer #3: Yes

Reviewer #4: Yes

Reviewer #5: Yes

Reviewer #6: Yes

Reviewer #1: Qualitative evaluation of a pilot mental health program for public safety personnel with

post-traumatic stress injury.

Manuscript Number PMEN-D-25-00363

This study examines a very welcome program in the field of mental health. Services for traumatized individuals are particularly rare. In order to highlight the urgency of such services, terms from psych traumatology should be introduced and discussed in this study (see comments on the discussion). Therapy services tailored to the needs of specific target groups are important for maintaining the ability to work and preventing chronicity. I therefore recommend clarifying even more clearly what would need to be adapted during implementation.

Introduction:

- I recommend briefly explaining the distinction between the terms PTSD and PTSI and why PTSI is used in the context of PSP.

- Line 74: How can the large range be explained?

- The symptoms of PTSI should be presented and explained why they can potentially render a person unable to work.

Methods:

- Why was no established qualitative method used as a basis, e.g., Qualitative Content Analysis (according to Mayring or Kuckartz)?

- How was the interview guide created? Deductively from the framework? Were inductive codes also generated from the text material?

- To what extent does the interview guide differ from the framework developed by Proctor et al.?

- Who created the interview guide? Was this reflected in a collegial exchange? Line 192: What exactly are the interviewers' areas of expertise?

- Were notes taken during the interviews?

- I recommend making the code system available in the appendix.

- Information on the following aspects is missing:

oWhat rules were used for recruitment? (Selective sampling? Maximum variance?)

oWhat are the demographic characteristics of the sample?

oWhat criteria were used to end recruitment?

- To increase transparency, I recommend attaching the COREQ checklist as a supplement: Tong A, Sainsbury P, Craig J. Consolidated criteria for reporting qualitative research (COREQ): a 32-item checklist for interviews and focus groups. International Journal for Quality in Health Care. 2007. Volume 19, Number 6: pp. 349 – 357

- When was the coding process completed? Was content saturation achieved? If not, please explain why and include this in the limitations section if applicable.

- To what extent were the quality criteria taken into account? Was countercoding used?

- Lines 186/187 are unclear: who are the “staff involved in the comprehensive assessment”?

- Line 199: Shouldn't it say ‘effectiveness’ instead of ‘acceptability’?

- The analysis process should be described in more detail: How was the text processed to increase the level of abstraction? Were memos created? Were inductive codes created?

Results:

- Line 212: Does a nurse belong to the PSP group?

- I recommend reporting which aspects were mentioned very frequently in the interviews.

Discussion:

- Overall, the discussion still lacks depth!

- At the beginning: a brief summary of the results.

- I miss key terms from psych traumatology that can be discussed very well based on the quotations, e.g.:

osecond-victim phenomenon

oresilience

obloc-building effect

oavoidance behavior

osafe space

- On the question of effectiveness: The timing of the survey is significant from a traumatological point of view because processing traumatic content takes time (several months!). This means that if the survey is conducted too soon after the intervention, it may not be possible to evaluate its effectiveness. In addition, the participants expressed that psychoeducation is an important aspect in dealing with the symptoms.

- It would be interesting to know whether the questionnaires were completed by the participants themselves or by the therapists (bias of socially desirable responses!?)

- Line 580: This is a recruitment bias and should be identified as such, as only those who are fundamentally open to therapy participate.

- Comorbidity: This low-threshold offer may not be suitable for taking comorbidities into account. However, it is still justified because it prevents post-traumatic stress from becoming chronic. My recommendation would be to emphasize prevention rather than therapy in the discussion.

- Formatting: Some of the quotations are not indented (e.g., line 392).

- The PSP quotes suggest that training peers who are familiar with the work environment and terminology can be useful in increasing acceptance.

- The social aspect could possibly be discussed: the public should be interested in the mental health of their security personnel. That is why such tailor-made programs are very valuable!

Conclusion:

- The weaknesses of the program are clearly evident from the quotations. A concluding presentation of the aspects that should be improved would be desirable.

Reviewer #2: Strengths:

1. The study fills a gap by comparing PSP and provider perspectives, which is rarely done.

2. The paper offers policy and programmatic implications for implementation of trauma-informed care.

3. The three organizing constructs (appropriateness, acceptability, effectiveness) are well-applied.

Areas for improvement:

1. The introduction describes PSP vulnerability well, but it reads more as an extended literature review than a conceptual framing. Consider adding a short paragraph near the end of the Introduction linking PSP treatment experiences to implementation science and cultural competence theories (e.g., Normalization Process Theory, or cultural safety frameworks).

2. Currently, “credibility” emerges inductively in findings but is not theoretically introduced—yet it becomes the linchpin concept. Introducing “credibility” as a theoretical construct (trust, legitimacy, cultural safety) earlier would improve coherence.

3. The methods mention that one interviewer had trauma-related expertise but not PSP lived experience. Reflexivity could be explicitly discussed e.g., how researcher background, gender, or institutional affiliation may have influenced interviews or interpretation.

4. Specify how saturation or theme stability was determined.

5. The analysis is rich but largely descriptive. Some findings could be elevated with modest interpretive commentary (e.g., the “abrupt discharge” theme could be tied to continuity of care literature or models of recovery capital; and “virtual delivery” could engage more critically with the literature on digital relationality and trauma therapy).

6. Clarify distinction between prevalence of PTSI and exposure to PPTEs to avoid conflating them.

7. The framing of “effectiveness” is somewhat ambiguous. It oscillates between program evaluation sense and participant perception. Clarify early that the study examines perceived effectiveness. Similarly, “formative evaluation” should be briefly defined and distinguished from summative evaluation.

8. There are several repeated citations (e.g., Haugen et al., 2012; Lewis-Schroeder et al., 2018 appear twice). Check numbering consistency after editing.

Reviewer #3: The cohort for this study is relatively small, with only 19 PSP participants; however, the study is clearly and concisely defined, and as is detailed in the submission, due to the lack of research in this field, it is highly valuable and relevant in terms of person-centered care and a defined need for more specialised interventions for individuals who are disproportionately affected by PTSI’s.

Throughout the submission, there is mention of a lack of involvement of PSP in study design and in integrating their perspectives in concept/study development. This heightened awareness of the importance of lived experience engagement in research is critical in understanding and respecting outcome limitations, and it is good to see it mentioned and brought to attention numerous times.

The use of participants' quotes is always a welcome addition and authentically supports the submission.

Some minor notes/ considerations:

In terms of effectiveness, I question the measure of this, as is stated in line 465 “These different conceptions of effectiveness meant that how the program was seen to support effectiveness varied considerably.” I would encourage further exploration into empowerment, self-efficacy, self-worth & connectedness.

Sustainability of the study is also unclear. It would be valuable to add what happens after this pilot and how ongoing support for participants could be integrated into future efforts.

Well done & thank you to all the authors for your contributions.

Reviewer #4: General Assessment

This is a timely and well-executed qualitative evaluation of a pilot mental health program for Public Safety Personnel (PSP). The manuscript is well-written, methodologically sound, and offers significant insights drawn from the dual perspectives of both service providers and PSP clients. The findings on the centrality of credibility, the need for a PSP-exclusive environment, and the importance of cultural competence among staff are valuable contributions to the field.

Strengths

1. Robust Methodology: The use of a qualitative descriptive design, guided by the Proctor et al. framework, is highly appropriate for this formative evaluation. The data analysis process, involving separate coding by two team members, reflects methodological rigor.

2. Novel Contribution: The study effectively addresses a gap in the literature by directly comparing the experiences and perspectives of treatment staff and the PSP they serve, a viewpoint often missing in prior research.

3. Rich Findings: The results are presented with depth and are well-supported by powerful participant quotes. Key themes—such as the validation found in peer groups, the need for program structure, and challenges with the discharge process—are clearly articulated and impactful.

4. Balanced Discussion: The discussion section effectively situates the findings within existing literature and thoughtfully acknowledges the study's limitations.

Minor Suggestions for Improvement

While the manuscript is strong, the following minor revisions could further enhance its impact:

1. Addressing Cumulative Trauma: A participant noted the program felt geared toward singular traumatic events rather than cumulative trauma. The Discussion section could be strengthened by exploring how future programs might better accommodate the cumulative nature of trauma prevalent among PSP.

2. Elaborating on Virtual Delivery Challenges: The paper highlights the trade-off between accessibility and connection in virtual therapy. It would be beneficial to discuss potential solutions, such as hybrid models or specific virtual facilitation techniques, to mitigate the limitations of online group sessions.

3. Refining Group Composition Recommendations: The finding that group composition (e.g., rank, gender) impacts engagement is critical. Highlighting this as a specific, actionable recommendation in the conclusion would be valuable for program designers.

4. Emphasizing Patient-Centered Discharge: The abruptness of the discharge process was a significant point of distress for participants. The authors' suggestion for a tapered transition is excellent and deserves greater emphasis as a concrete recommendation to prevent an erosion of progress.

Reviewer #5: This is a generally well-written exposition of a pilot MH program to treat public safety personnel (PSP) with post-traumatic stress injury. It is very detailed in its presentation of narrative interviews of PSP and the persons trained to conduct their therapy program. While overall an interesting article, and useful as an early report on specialty treatment programs for first responders, there are areas that are too verbose and other areas lacking in sufficient detail, as follows.

The concept of post-traumatic stress injury (PTSI), as used in the title and throughout the MS, is never defined. The authors never explicitly state that it is different from syndromal PTSD, and the PCL-5 rating scale the authors used does have a threshold for civilian PTSD. So, how many of the PSP participants had full, syndromal PTSD and how many did not (e.g., subsyndromal PTSD – Other Specified Trauma- and Stressor-Related Disorder in DSM-5)? How did they differ in traumatic event exposure, treatment response, etc.?

Abstract: The last sentence, lines 48-51 is vague – what were some of the conflicting goals? If the PSP made no progress by established metrics but still were better able to manage their PTSI (a vague statement in itself), then were the metrics off the mark? Better to remove this sentence and leave it for the Discussion, where the details can be presented.

Introduction: Line 59: “disproportionately” – vs whom? Suggest eliminating this sentence and putting ref 5 with previous sentence. Line 60: “considerable barriers to treatment” – a provocative statement; give some examples of a “considerable barrier.” Line 67: PSP were on occupational leave for PTSI – was this syndromal PTSD? See comments in paragraph 2 above. Note that line 95 refers to PTSD, which is mentioned a few more times throughout the MS. Lines 97-123: These paragraphs are redundant to earlier sections of the Introduction and can be integrated with the earlier sections. Line 126: Move (FRMH) up to line 125 after First Responder Mental Health.

Methods: Line 143: Participants (plural). Line 150: Delete initial “the.” Line 166: Suggest adding “residential” after “main.”

Results: Of those invited to participate, what % do the 11 providers and 19 PSP represent? How do they compare demographically to the total invited sample? Is there some selection bias here? And, what were the psychotropic medications the PSP were taking? A table indicating the demographics of both groups, including PSP comorbidities and medications, would be helpful. Line 219: “…challenges for PSP with complex needs…” is a provocative but vague statement. What kind of challenges and what kind of complex needs?

Overall, the Results section is wordy. The participant quotes are useful, but there are a lot of them, and many contain redundant concepts. The quotes thus could be condensed. Following are three examples; please review the others and condense as much as possible.

Lines 244-249: “Every single one of us in the group said…having our feelings and experiences validated by hearing someone else describe events, or feelings, or complete inabilities that we also experienced made us feel we weren't alone…and other people are out there going through this…This is something that happened to me, this isn't who I am. (PSP027, corrections worker)”

Lines 295-299: “In group therapy, when it happens in person, we don’t have control over…when they’re exchanging numbers, they’re meeting outside for coffee, maybe they build friendships…a lot of these folks are…so isolated. They don’t have friends, and…they cannot talk to anyone who can resonate with them... (Clin014)”

Lines 326-331: “I, very early on I realized that there was going to be a problem…because it wasn't a singular incident that brought this on, it was accumulation. And…all the work…and even the PTSD questionnaires, didn't reflect that, it was like, all the questions were geared to singular events… (PSP026, paramedic)”

Lines 367-369: Have you considered having the staff go on calls with the PSPs to have first-hand knowledge of the challenges of being a particular type of PSP? Lines 375-379: Please detail the explanations of confidentiality that were given to the PSPs at the beginning of the program. Line 464: “…different conceptions of effectiveness…” is a vague statement. What were the different conceptions and the resulting participant perceptions of effectiveness. Give some examples, please.

Overall, the Results section is very long. In addition to condensing the many quotes, please consider condensing the text as well.

Discussion: Line 531: Delete “Appropriate” and start sentence with “Treatment”. (Appropriate vs what? Inappropriate?) Lines 553-554: How many PSPs returned to work? This might be added to the table and referred to in this part of the Discussion. Lines 599-600: The PCL-5 is a subject-rated questionnaire and can be biased depending on the circumstances (e.g., over-scored by veterans seeking service-related monetary compensation), so that clinician-rated questionnaires; e.g., the CAPS, might serve the program better. Perhaps consider this under Future Research.

In summary, this is an interesting and potentially useful article and can be strengthened considerably by attention to the above comments and suggestions, particularly condensing the MS and including a table.

Reviewer #6: The manuscript offers a valuable perspective on mental health programs for public safety personnel with post-traumatic stress injury. It is well-detailed, technically sound, and demonstrates potential for replicability. However, the authors should consider addressing the following points to further strengthen the manuscript:

Clarification of Line 78–79

The authors should explicitly list the specific events being referenced in the statement, “Members of the general public are exposed to only three of those events.”

Citation Needed (Line 92–94)

A supporting citation is required for the claim made in this section to enhance credibility and academic rigor.

Recruitment and Sampling Details (Line 178)

More detailed information on the participant recruitment process and sampling methodology is recommended for improved transparency.

Theoretical Framework Visualization (Line 157)

If possible, a diagrammatic representation of the theoretical framework would enhance reader comprehension.

Clarification on Data Collection Tools

It would improve clarity if the authors explicitly mention that two separate data collection tools were used, describe each tool, indicate whether they were adapted from previous studies, and state whether they were validated.

The authors should ensure that the conclusion remains objective and is explicitly aligned with the stated objectives of the study. The final statements should reflect the study’s findings without overgeneralization or speculative claims.

**Do you want your identity to be public for this peer review?** For information about this choice, including consent withdrawal, please see our Privacy Policy

Reviewer #1: **Yes:**  Antina Beutel

Reviewer #2: No

Reviewer #3: **Yes:**  Sandra Ferreira

Reviewer #4: No

Reviewer #5: No

Reviewer #6: **Yes:**  Gerald Obinna Ozota

---

## [Editor Report · Decision Letter 1]

21 Dec 2025

Qualitative evaluation of a pilot mental health program for public safety personnel with post-traumatic stress disorder

PMEN-D-25-00363R1

Dear Dr. Wodchis,

We are pleased to inform you that your manuscript 'Qualitative evaluation of a pilot mental health program for public safety personnel with post-traumatic stress disorder' has been provisionally accepted for publication in PLOS Mental Health.

Best regards,

Lambert Zixin Li, Ph.D.

Academic Editor

PLOS Mental Health

Dear Authors,

Thank you for your revision. I am pleased to accept the manuscript for publication.

Sincerely,

Lambert Zixin Li, PhD